# The Ocular Microbiome: Micro-Steps Towards Macro-Shift in Targeted Treatment? A Comprehensive Review

**DOI:** 10.3390/microorganisms12112232

**Published:** 2024-11-04

**Authors:** Ewelina Trojacka, Justyna Izdebska, Jacek Szaflik, J. Przybek-Skrzypecka

**Affiliations:** 1SPKSO Ophthalmic University Hospital in Warsaw, 03-709 Warsaw, Poland; ewelinatrojacka@gmail.com (E.T.); justyna.izdebska@wum.edu.pl (J.I.); jacek.szaflik@wum.edu.pl (J.S.); 2Department of Ophthalmology, Medical University of Warsaw, 03-709 Warsaw, Poland

**Keywords:** ocular microbiome, ocular microbiota, ocular dysbiosis, keratitis

## Abstract

A healthy ocular surface is inhabited by microorganisms that constitute the ocular microbiome. The core of the ocular microbiome is still a subject of debate. Numerous culture-dependent and gene sequencing studies have revealed the composition of the ocular microbiome. There was a confirmed correlation between the ocular microbiome and ocular surface homeostasis as well as between ocular dysbiosis and pathologies such as blepharitis, microbial keratitis, and conjunctivitis. However, the role of the ocular microbiome in the pathogenesis and treatment of ocular surface diseases remains unclear. This article reviews available data on the ocular microbiome and microbiota, their role in maintaining ocular homeostasis, and the impact of dysbiosis on several ophthalmic disorders. Moreover, we aimed to discuss potential treatment targets within the ocular microbiota.

## 1. Introduction

The National Institutes of Health initiated the Human Microbiome Project (HMP) in 2008. The goal was to map the microbial species that cohabit the human body in five main areas: oral cavity, respiratory tract, skin, digestive tract, and vagina [1,2]. The project revealed that the microbiota cohabitating the human body contains around 8 million genes of bacteria, viruses, and eucaryota [1,2]. Notably, the results demonstrated that the microbes colonizing several niches in the human body vary individually. These organisms participate in the maintenance of well-being but also in the development of pathologies [3,4,5].

The ocular surface was not explored within the Human Microbiome Project, but the results raised the question of whether mucosal membranes, such as those in the ocular surface, also possess resident microbiota. It appears that characterizing the ocular microbiome is more challenging.

The ocular surface is not homogeneous. Many microhabitats exhibit unique characteristics and distinct microbial communities [6,7]. Furthermore, the ocular surface is in close proximity to the external environment, which increases the risk of contaminating species disturbing the healthy microbiome [8]. Another issue is the commensal microflora on the ocular surface steadily being exposed to defense reactions [9]. Despite these challenges, studies have identified an ocular surface microbiota that represents a persistent and stable consortium of viable organisms on the ocular surface [10]. The number of bacteria on the ocular surface is estimated to be much lower than that on other mucosal surfaces, approximately 150-fold less than that on the face skin or cheek mucosa [11]. Despite the low bacterial concentration, there is a high impact on ocular diseases. The ocular microbiome plays a role in the pathogenesis of ocular surface disorders. Nevertheless, some eye diseases are a consequence of gut microflora dysbiosis, providing evidence for the existence of the gut–eye axis [12].

The aim of this study was to determine the healthy ocular microbiota and its role in maintaining ocular homeostasis and in the development of ocular pathologies. In addition, we investigated potential treatment targets within the ocular microbiome.

## 2. Investigation Methods of the Ocular Surface Microbiota

The first studies documenting the presence of microbes on the ocular surface date back to 1930, when only conventional culture techniques were used to identify the microorganisms [13]. Over the years, improved and sensitive research techniques have made it possible to obtain more precise data. High-tech-mode gene sequencing has become the gold standard for studies on the ocular surface microbiome. The most reliable diagnostic methods are 16SrRNA gene-based sequencing and whole-metagenome shotgun sequencing [8,14]. Conventional culture techniques are limited in detecting the ocular surface microbiota owing to the specific growth requirements of individual microorganisms and their low sensitivity [15,16,17,18]. However, each diagnostic method has its own limitations. Despite providing high sensitivity and detecting every gene on the ocular surface, gene sequencing techniques cannot differentiate between viable residents of the ocular surface and contaminants [19,20,21,22,23,24,25]. Table 1 depicts advantages and disadvantages of the three aforementioned methods of the ocular microbiome examination.

## 3. Ocular Microbiome or Ocular Microbiota?

Two non-synonymous communities were described based on the aforementioned diagnostic methods: the ocular microbiome and the ocular microbiota [14]. The ocular microbiome is composed of genetic material comprising bacteria, fungi, viruses, protozoa, and eukaryotes on the ocular surface, whereas the ocular microbiota is a community of organisms that colonize the ocular surface [20]. Defining the core ocular microbiota, a set of microbial taxa characteristic of a specific host or environment, remains a subject of debate [37].

## 4. Ocular Microbiota Characteristics

One of the first studies to investigate the composition of the ocular microbiota was the Ocular Microbiome Project, initiated in 2009 by scientists at the Bascom Palmer Eye Institute [38].

Samples obtained from four individuals were analyzed using 16SrRNA gene-based sequencing. The initial results identified 59 distinct bacterial genera, of which 12 were ubiquitous among all examined subjects. However, the small sample size limited the analysis [38]. Since then, many trials analyzing the ocular surface microbiota have been undertaken, using different diagnostic methods and research protocols, which makes it difficult to summarize their results and characterize the exact composition of the ocular surface microbiota. Varying results were obtained depending on the method used for collecting and investigating sample. “Light pressure wiping” enables detecting some genera, namely, *Rothia*, *Herbaspirillum*, *Leptothrichia*, and *Rhizobium*, while reducing the detection of others, i.e., *Firmicutes* (*Staphylococci*), *Actinobacteria* (*Corynebacterium* spp.), and *Proteobacteria*. “Strong pressure wiping” resulted in a higher abundance of *Proteobacteria*, *Bradyrhizobium*, *Delftia*, and *Sphingomonas* on the conjunctival epithelium. The microbial fraction may be easily washed away from the ocular surface by mucus; therefore, deep pressure is recommended over scraping when studying ocular surface microorganisms [39].

Culture-based methods revealed the presence of coagulase-negative *Staphylococci*, *Propionibacterium*, *Corynebacterium*, *Staphylococcus aureus*, and *Streptococcus* as the most abundant genera on the ocular surface [39,40,41,42]. 16SrRNA gene-based sequencing indicated a slightly different composition—the dominant phyla were *Actinobacteria* (53%), *Proteobacteria* (39%), and *Firmicutes* (8%), followed by *Corynebacterium*, *Acinetobacter*, *Pseudomonas*, *Staphylococcus*, *Propionibacterium*, and *Streptococcus* [8,43]. Table 2 presents data on the composition of the ocular surface microbiota.

The advent of modern sequencing technologies has enabled extensive microbiome characterization. However, 16SrRNA gene sequencing—the most commonly used sequencing technology in microbiome research to date—aims to target only bacteria, thus omitting other kingdoms. Whole-metagenome shotgun sequencing is an emerging sequencing technology that, compared to 16SrRNA gene sequencing, provides more comprehensive details of the taxonomic composition and offers the possibility to study functional profiles of the microbiome. It increases taxonomic resolution and provides deeper genomic information [24,33]. Due to its high precision, studies using whole-metagenome shotgun sequencing appear to provide the most accurate results.

Based on these technologies, it was established that the ocular surface microbiome comprises bacteria, viruses, and eukaryotes. Bacteria were detected in 73–98% of the analyzed samples and were thus the dominant group of microorganisms in the ocular surface microbiota. Viruses were detected in 1–7% and eukaryotes with dominant fungi in 0.02–20% [44,45,46,53,54].

Interactions between components of ocular microbiome are crucial for the sustainability of various ecosystems and maintaining the homeostasis of the ocular surface. Viral and bacterial interactions create the possibility for the modulation of antiviral immune response and viral infectivity. Bacteria-infecting viruses (bacteriophages) are one of the main regulators of bacterial population density and distribution [55,56,57,58]. Moreover, bacterial composition on the ocular surface may influence susceptibility to fungal infection. It was discovered that individuals developing fungal keratitis have an altered bacterial composition not only on the surface of the affected eye but also in the fellow non-affected eye when compared to healthy subjects’ eyes [59].

## 5. Factors Modifying the Composition of the Ocular Surface Microbiota

The composition of the ocular surface microbiota may be modified by age, sex, microhabitat, contact lens use, and several topical treatments [14,60]. Moreover, there is a direct link between alterations in ocular surface microbiota and ocular/general diseases, which may be both a cause and a consequence of dysbiosis.

The composition of the ocular surface microbiota changes from birth to adulthood and remains relatively stable throughout life. The neonatal conjunctiva has a higher level of positive cultures and a greater diversity of species than other stages of the human life cycle.

Coagulase-negative *Staphylococcus* and *Propionibacterium* dominate the ocular microflora of neonates delivered naturally; however, microbes similar to those in the cervix are also commonly detected. After two days of life, fewer microbes are isolated from the conjunctiva, dominated by *S. epidermidis*, *E. coli*, and *S. aureus* [8,61,62,63]. The ocular surface microbiota gradually becomes increasingly similar in composition to that of the adult. Notably, the general pediatric population presents with more numerous species, including dominant aerobic cocci and *Propionibacterium* [64]. An increased proportion of anaerobic cocci and *Corynebacterium* becomes more frequent with age. The most prominent diversity of ocular microbiota was observed in older patients, when 28-to-84-year-olds were studied [47].

Sex also influences the ocular microbiota. Based on available data, the female ocular surface microbiota is characterized by a lower abundance of *P. aeruginosa* and *S. epidermidis* and a higher abundance of *E. coli* [47].

## 6. Microhabitats of the Ocular Microbiota

There are various microhabitats on the ocular surface with significantly different resident microbial communities. Differences have been confirmed, especially between the lid margins or skin, which has the highest abundance of bacteria, and the bulbar conjunctiva, with the lowest concentration. The composition of the ocular microflora also differs in particular microhabitats; for instance, *P. aeruginosa* is distributed at a higher relative abundance in the conjunctiva and eyelid tissue than in the skin. Notably, no differences were found in the bacterial communities between the limbus and the fornix [8].

## 7. Ocular Microbiome in Contact Lens Wearers

Use of contact lenses affects the ocular microflora, resulting in dysbiosis and a higher risk of ocular infection [65]. This influence is affected by the lens material, mode of lens use, and user age [11]. The results obtained by Shin et al. in a group of contact lens wearers (using 16S rRNA gene-based sequencing) showed that dry conjunctival swabs from lens wearers had more variable and skin-like bacterial community structures, with a higher abundance of *Methylobacterium*, *Lactobacillus*, *Acinetobacter*, and *Pseudomonas* and a lower abundance of *Haemophilus*, *Streptococcus*, *Staphylococcus*, and *Corynebacterium*, in comparison with those who do not use contact lenses [65].

There were differences in the bacterial abundance between soft contact lens wearers and orthokeratology lens wearers [66]. *Bacillus*, *Tatumella*, and *Lactobacillus* plethora was reduced in orthokeratology lens users compared to non-CL-users. The *Delftia* abundance decreased, whereas *Elizabethkingia* levels increased in soft contact lens wearers compared with non-wearers. The difference between the soft contact lens wearer and non-wearer groups was less prominent than that in the orthokeratology group [66].

Age also influences the microbiota composition in CL wearers. In the adult population of soft contact lens wearers, the ocular microflora is more abundant, whereas no differences in microbiota composition were observed in children aged 8–14 years using 2-hydroxyethyl methacrylate (HEMA)-based soft contact lenses for more than two years [67]. Zhang et al. reported that the relative quantity of the microbial community in the conjunctival sac of myopic children aged 8–15 years who wore orthokeratology lenses for 12–13 months may be altered. Differential genera were identified with 16S rRNA gene-based sequencing as follows: *Muribaculaceae*, unclassified; *Blautia parasutterella* and *Muribaculum*, more abundant in the orthokeratology group; and *Brevundimonas*, *Acinetobacter*, *Proteus*, and *Agathobacter*, more abundant in the non-wearer group. Moreover, changes in microbiota metabolism were identified in the orthokeratology contact lens-wearer group [68].

An increase in the number of bacteria isolated from the conjunctiva and lids during daily lens wear has been reported. However, extended lens use was associated with an alteration in the types of microorganisms, and more Gram-negative bacteria were isolated [69].

The extended wearing of HEMA-hydrogel and silicone contact lenses increased the microbial expansion within the eyelid margin and bulbar conjunctiva [70]. Moreover, contact lenses also alter the balanced composition of the microbiome, mainly by increasing Gram-positive concentrations in the lower eyelid margin area and decreasing concentrations in the upper conjunctival regions [71].

Notably, contact lens-associated red eye may result from ocular microbiota alterations induced by contact lens use. The data also showed that *H. influenzae* was more frequently isolated in eyes with symptomatic subepithelial corneal infiltrates than in those with no ocular surface changes [71].

Furthermore, the presence of Gram-positive microorganisms on contact lenses, such as *Corynebacterium* spp. and coagulase-negative *Staphylococci*, is associated with higher susceptibility to the development of peripheral corneal ulcers. In turn, the presence of Gram-negative microorganisms on contact lenses promotes the development of contact lens-associated red eye [71]. However, in cases of contact lens-associated red eye, soft contact lenses are the main contributor to the development of dysbiosis [71].

## 8. Ocular Microbiome in Topical Ophthalmic Therapy

Topical ocular treatment alters the microbiome. Ubiquitously applied artificial tears lower the culture-positive rate of bacteria in the conjunctival sac, mainly because of physical wash-out [40]. Interestingly, there was no change in the composition of the ocular surface microbiota after lubricant instillation [40].

Zhou et al. reported that the use of carboxymethylcellulose for seven days did not affect ocular surface microbiome diversity but several modifications in microbial concentrations were noted: an increase in *Acinetobacteria* and a decrease in *Bacteroides* and *Firmicutes* [72]. Significant changes in the bacterial community were reported by Zhong et al. in samples collected from patients using 0.3% sodium hyaluronate eye drops with or without the preservative benzalkonium chloride (BAK) four times a day for 2 weeks in both eyes. Decreased relative abundances of *Flavobacterium caeni* and *Deinococcus antarcticus* were observed in both groups, independent of the presence of BAK [73].

Antibiotic drops differ in their effects on the ocular microbiome. In general, dysbiosis caused by antibiotics disturbs ocular homeostasis, enabling pathogens to invade the ocular surface and, finally, select antibiotic-resistant strains [74]. The dysbiotic impact of topical azitromycin, gatifloxacin, moxifloxacin, and ofloxacin on ocular microflora was confirmed [74,75]. The most prominent disruption was observed in *S. epidermidis* and *S. aureus* colonies in the ocular surface microbiota [74]. Fewer Gram-positive bacteria were detected in the ocular microflora during therapy with topical tobramycin and moxifloxacin [75]. Other topical fluoroquinolones are also associated with a reduction in the number of Gram-negative microorganisms [74].

Topical glaucoma therapy also alters the composition of the ocular microbiota [76]. A higher concentration of Gram-negative and anaerobic bacteria has been reported compared to healthy controls. The differences were associated with a decreased tear film meniscus and were observed in both treated eyes and contralateral untreated eyes. However, based on the aforementioned study, it is impossible to determine whether the microbial changes were related to the drops used to treat glaucoma or to the disease itself. Moreover, alterations in the microbiomes of both eyes (treated and untreated) suggested that the ocular surface microbiome of the two paired eyes may act as a singular microbial ecosystem [76].

Honda et al. found a similarly high abundance of Gram-negative bacteria in the ocular surface microbiome after topical anti-glaucomatous therapy, although the study was restricted to culturable microbes [77]. It is also suggested that if eye drops act on the ocular surface microbiome, the active medication ingredient may not be responsible for inducing the changes but, rather, ingredients common to all drops, such as preservatives [76]. Benzalkonium chloride (BAK) is the most frequently used preservative in topical glaucoma treatment. At low concentrations, it primarily inhibits the growth of Gram-positive organisms, whereas at higher concentrations, it inhibits the growth of Gram-negative organisms [78,79]. The presence of BAK persists on the ocular surface with a half-life of 20 h in corneal epithelial tissues and 11 h in deeper conjunctival layers, with detectable concentrations up to one week after instillation [80]. This is consistent with the hypothesis that chronic daily exposure to low concentrations of BAK in preserved ophthalmic drops can generate persistent changes in the ocular surface microbiome [76]. Pathologists have found that tear film disruption by BAK creates a local hypoxic ecological niche that preferentially selects Gram-negative anaerobes [81]. However, in a recent study, Priluck et al. found that in a group comprising glaucoma patients solely, BAK had no effects on the ocular surface microbiome [82].

## 9. Role of the Ocular Microbiome and Its Alterations in Ocular Diseases

Microorganisms residing on the ocular surface communicate with epithelial and immune cells and coordinate different functions aimed at the maintenance of homeostasis and local well-being. They preserve barrier function, inhibit apoptosis and inflammation, and accelerate wound healing. Moreover, they protect their niches from pathogenic invasion via competition and interact with immune system [83].

Commensals and potential pathogens are recognized by the immune system through interactions between antigens and receptors, called pattern recognition receptors (PRRs), of which Toll-Like Receptors (TLRs) are the most important. These reactions usually result in rapid inflammatory response, but not in the case of ocular surface microbiome components [84,85,86]. The corneal epithelium expresses TLR-2 and TLR-4 not on the cellular surface but at the intracellular level, which, in comparison to normal surface expression, allows a condition of “immune silence” on the ocular surface that prevents unnecessary inflammatory responses against organisms belonging to the ocular microbiome [44,46,87].

In addition, there are other hypotheses explaining this unordinary relationship between ocular microbiome and immune system. Some authors suggest that membrane TLRs may be inactive or not expressed at the protein level, whereas others indicate the existence of several molecules that can block TLR signaling [88,89,90]. This allows for the presence of commensal microorganisms on the ocular surface and suggests that the ocular surface microbiome is in a symbiotic relationship with the immune system and plays a role in the education, function, and induction of the immune system [91,92].

Microbiome disturbances are associated with ocular surface pathologies, which may be both a consequence and a cause of dysbiosis. Table 3 summarizes the altered composition of physiological microbiota in different ocular disorders.

### 9.1. Blepharitis and Meibomian Gland Dysfunction

An increase in the relative abundance of *Staphylococcus*, *Streptophyta*, *Corynebacterium*, and *Enhydrobacter* genera, along with a decrease in *Propionibacterium*, was observed in patients with blepharitis [93].

Individuals with moderate Meibomian gland dysfunction had the lowest concentration of *Staphylococcus*, whereas those with severe Meibomian gland dysfunction had a higher abundance of *P. acnes* in the ocular surface microflora [94].

### 9.2. Dry Eye Syndrome

Ocular microbiome dysbiosis is associated with inflammation and may lead to the development of symptoms and the progression of dry eye disease. Increased concentrations of Gram-positive bacteria, especially coagulase-negative *Staphylococci*, *S. aureus*, and *Corynebacterium*, as well as pathogens, such as *Rhodococcus* and *K. oxytoca*, are a consequence of dry eye syndrome [95,96].

Dry eye syndrome is also associated with changes in gut microbiota. Increased pro-inflammatory bacteria number and decreased short fatty acid-related bacterial genera that produce anti-inflammatory effects, modify the homeostasis of the ocular microenvironment. In relation to this, fecal microbiota transplantation or probiotic intervention alleviate signs of inflammation on the ocular surface of dry eye animal models [97].

There is a hypothesis that antimicrobial components of the tear film may influence the ocular surface microbiome and vice versa. Based on this hypothesis, a disturbed tear film composition could favor the dysbiosis of the ocular surface microbiome, lower protection against pathogens, and lead to diseases [44,98]. In addition, it is hypothesized that the ocular microbiota may influence the metabolism of amino acids which are normally in the tear film and play a role in maintaining ocular surface homeostasis [46,99,100].

### 9.3. Trachoma

Ocular surface microbiome alterations in individuals with trachoma with conjunctival scarring are characterized by decreased bacterial diversity and the overgrowth of *Corynebacterium* and *Streptococcus* species [101].

### 9.4. Allergic Conjunctivitis

There is a negative correlation between the severity of allergic conjunctivitis and the decreased diversity of the ocular surface microflora [102]. Moreover, in individuals with allergic conjunctivitis, the bacterial community in the nasal mucosa was more similar to that in the conjunctiva [102].

### 9.5. Infectious Keratitis

A healthy ocular microbiota protects the ocular surface from pathogenic invasion. It plays a preventive role mainly in the development of infectious keratitis [12].

The ocular microbiota constantly stimulates immunological effectors and creates specific competition. Such correlation was confirmed in Swiss Webster mice, which were usually resistant to *P. aeruginosa*-induced keratitis. After the alteration of the microbiome, they become sensitive to this pathogen. Immunity was subsequently re-established by colonizing the ocular surface with coagulase-negative *Staphylococci* [103].

A similar protective effect of *Corynebactrium mastits* in colonizing human skin has been suggested. It was hypothesized that these bacteria could stimulate the production of interleukin-17 by conjunctival T lymphocytes, thereby allowing the recruitment of more neutrophils [104].

Individuals suffering from keratitis present with an overgrowth of potentially pathogenic genera, such as *Pseudomonas* and *Acinetobacter* [102].

### 9.6. Sjögren’s Syndrome

Sjögren’s syndrome is a specific subtype of dry eye syndrome. This indicates a connection between the gut microbiota and systemic immune reactions in distant body areas. There is a suggested correlation between severe dry eye in Sjögren’s syndrome and the dysbiotic intestinal microbiome [105].

Compared to healthy subjects, patients with Sjögren’s syndrome present with a reduced diversity of gut microbiota [106,107,108]. A two-sample Mendelian randomization study provided evidence for either positive or negative causal effects of gut microbiota composition and related genes on Sjögren’s syndrome risk. *Fusicatenibacter* and *Ruminiclostridium 9* were positively correlated with the risk of Sjögren’s syndrome, whereas *Subdoligranulum*, *Butyricicoccus*, and *Lachnospiraceae* were negatively correlated with the risk of Sjögren’s syndrome [109].

Moreover, butyrate-producing bacteria, including *Faecalibacterium prausnitzii*, *Bacteroides fragilis*, *Lachnoclostridium*, *Roseburia*, *Lachnospira*, and *Ruminococcus*, were substantially reduced in patients with Sjögren’s syndrome [110,111,112].

### 9.7. Uveitis

The gut microbiota may play a role in the development of uveitis, which has been proven in animal models of uveitis [113]. The transfer of the gut microbiota from patients with Behçet disease and Vogt–Koyanagi–Harada disease has resulted in a significant exacerbation of the disease in experimental autoimmune uveitis (EAU) mice [113,114]. Furthermore, oral antibiotic therapy attenuated ocular inflammation in EAU mice, in contrast to intraperitoneal administration [115].

It has been suggested that microbes and their metabolites can function as antigen mimics and activate peripheral T cells that cross the blood–retinal barrier and cause inflammation in the eye [116]. However, it was confirmed that propionate produced by gut bacteria increases Treg cells in the intestinal lymphocyte population at the early stage of EAU and promotes the maintenance of structural stability in the intestine [117,118,119]. Moreover, a large number of microbial metabolic peptides can bind the HLA B27 molecule and induce immune responses in target organs such as the eye [120]. Taken together, a correlation between gut microbes or their metabolites and the development of systemic immune response is suggested.

### 9.8. Age-Related Macular Degeneration

Gut microbiota may be involved in age-related macular degeneration (AMD) pathogenesis.

The pathophysiology of AMD involves the recruitment of microglia and macrophages in the subretinal and choroidal areas, mast cells, and RPE immune activation [121].

Intestinal dysbiosis has been described in individuals with advanced AMD compared to healthy adults. Differences were observed in the bacterial genera *Anaerotruncus*, *Oscillibacter*, *Ruminococcus torques*, and *Eubacterium ventriosum* [122]. Individuals with AMD had lower abundances of *Oscillospira*, *Blautia*, and *Dorea* [123,124].

Animal models of AMD have confirmed a correlation between the gut microbiota and inflammation during AMD progression. Mice fed a high-fat diet show intestinal dysbiosis and a two-fold increase in the number of microglia and macrophages within the local choroidal neovascularization (CNV) lesions. The eradication of the gut microbiota with neomycin reduces inflammation [122].

Moreover, mice fed with heat-killed *Lactobacillus paracasei* exhibit alteration in the *Firmicutes/Bacteroides* ratio and a decrease in retinal inflammation and ganglion cell loss [123].

The proven efficacy of oral antioxidative supplements in slowing AMD progression may be associated with the gut microbiota.

Mice with CNV that were fed a high-fat diet after a course of oral antibiotics showed a reduction in CNV and an improvement in microbiome composition [122]. The high-sugar diet, high-fat diet, and long-chain fatty acids used in murine models of AMD have also shown similar alterations in the intestinal microbiome. However, a low-sugar diet had no influence on retinal inflammation and the gut microbiota [124,125,126].

Despite many studies, it remains unclear whether a correlation exists between the gut microbiota and AMD.

According to a complex and still not fully recognized etiopathogenesis, many efforts are undertaken to find innovative effective treatment modality. Recent in vivo studies reported the strongly antioxidative, anti-inflammatory, and antiangiogenic efficacy of a nanotherapeutic formulation consisting of poly-Ɛ-capronolactone (PCL), cell-penetrating peptide (CPP), resveratrol, and metformine (R@PCL-T/M Nps) in a rat model of AMD. The authors suggest that a single-dose intravitreal injection of R@PCL-T/M Nps, due to the good biocompatibility of the nanotherapeutic formulation and the sustained release of resveratrol and metformin, could mitigate the disease progression for 56 days [127].

Nanoparticle-based therapies, as all new treatment modalities, despite their effectiveness, may have an influence on the ocular surface cells and microbiome, which calls for further research into such influence and optimized treatment strategies [128].

### 9.9. Glaucoma

Gut and oral microbiota dysbiosis plays a crucial role in the onset of neurological disorders and progressive neuronal loss [129]

Chen et al. described the induction of autoreactive T cells, which were previously pre-sensitized by symbiotic microbiota, to infiltrate the retina in response to a transient increase in intraocular pressure [130]. The dysfunction of retinal ganglion cells and axons, leading to irreversible damage due to elevated intraocular pressure, is characteristic of glaucoma pathogenesis.

Several studies have shown a potential correlation between the gut microbiota and glaucoma.

Differences in the composition of the gut microbiota and the serum metabolites have been confirmed between patients with glaucoma and healthy controls. Moreover, enhanced oral bacterial load is associated with an increased risk of developing glaucoma [131,132,133].

Studies in rats have resulted in butyrate, one of the gut microbiota metabolites, lowering the intraocular pressure independent of blood pressure changes [134]. Furthermore, increased levels of trimethylamine, a uremic toxin produced by the gut microbiota, have been detected in the aqueous humor of patients with glaucoma [135].

Specific memory T cells have been detected in murine models of glaucoma using heat shock proteins (HSPs) [130]. They were previously induced by intestinal commensals [104]. An elevated level of serum HSP27-specific antibodies was confirmed in experimental models [99]. Moreover, in human glaucoma, HSP27 and HSP60 have been found to be upregulated on retinal ganglion cells [130].

Based on these findings, a role of gut bacteria in the induction of glaucomatous damage has been suggested. Gut microbes stimulate CD4^+^ lymphocytes, which are able to escape the intestine and enter the eye through the damaged blood–retina barrier, causing neurodegeneration through cross-reaction with HSP-expressing retinal ganglion cells [136].

The exact role of the gut microbiota in the pathogenesis of glaucoma requires further investigation.

## 10. Gut–Eye Axis: Correlation Between Gut Microbiota Dysbiosis and Ophthalmic Diseases

The microbiota inhabiting the human gut accounts for more than 100 trillion bacteria, 70% of which occupy the large intestine [137,138]. A large body of evidence suggests that the gut microbiota is linked to the eye through the gut–eye axis. A disturbed balance of the gut microbiota leads to the translocation of pathogenic microbes through a compromised epithelial barrier and the alteration of the B cell and T cell populations, which results in systemic inflammation [137].

Many diseases with serious ophthalmic sequelae, including diabetes, multiple sclerosis, rheumatoid arthritis, and neurodegenerative diseases, are associated with disturbances in the gut microbiota [139,140,141,142,143,144]. According to a study published in 2023 by the *European Association for Predictive, Preventive, and Personalized Medicine Journal*, gut bacteria may influence ocular disorders [145]. There are hypothesized mechanisms that enable the gut microbiota and their metabolites to participate in the pathogenesis of ocular diseases, despite ocular sterility. Their main features include molecular mimicry and modified immunological pathways [116].

## 11. Ocular Microbiome in Therapeutic Strategies

Innovative therapies associated with the ocular surface microbiome and their roles in the pathogenesis of many ocular surface diseases are emerging. Anti-inflammatory, antimicrobial, and antihistamine therapeutics currently used for ocular surface disorders have numerous side effects and disturb ocular microbiome homeostasis, paradoxically stimulating the underlying pathology.

Alternatively, probiotics and prebiotics can be used to revitalize and stabilize effects on the ocular microflora.

A pilot study by Chisari et al. on a group of patients with dry eye syndrome showed statistically significant improvement in tear film stability and a lower abundance of *S. aureus* in the ocular surface microbiota after 30 days of treatment with oral probiotic supplementation [146].

Similar results were obtained after 4 and 8 weeks of combined therapy with pre-probiotics [147].

Kim et al. investigated the use of IRT-5 (a combination of *Lactobacillus casei*, *Lactobacillus acidophilus*, *Lactobacillus reuteri*, *B-bifidum*, and *Streptococcus thermophiles*) in dry eye syndrome therapy. Decreased symptoms of dry eye associated with autoimmune disorders were observed due to autoreactive T lymphocyte attenuation [148].

The effectiveness of topical 0.05% quercetin and resveratrol as probiotics was evaluated in a mouse model of dry eye. This treatment increased the tear film volume, improved corneal condition, and reduced ocular surface inflammation [149].

## 12. Conclusions

The ocular surface microbiome has gained increasing attention due to its potential role in the pathogenesis and treatment of various ocular diseases. Inflammatory processes are central to many of these conditions, and the microbiota, as a potential key modulator of inflammation, might become a promising target for novel ophthalmic therapies. Notably, the gut–eye axis may influence the progression of certain ocular diseases by altering the gut microbiota, suggesting that interventions targeting the microbiome could have therapeutic effects beyond the gastrointestinal system, the ocular system included.

Emerging evidence indicates that oral probiotics, prebiotics, and fecal microbiota transplantation, which are currently employed in treating gastrointestinal disorders, may also hold therapeutic potential for ocular pathologies. However, further research is necessary to evaluate their efficacy and safety in the context of ophthalmic treatments. Understanding the complex interactions between the microbiome and ocular health could pave the way for new diagnostic and therapeutic approaches in managing common eye diseases.

## Figures and Tables

**Table 1 microorganisms-12-02232-t001:** Summary of advantages and disadvantages of methods used in ocular microbiome exploration [13,15,16,17,18,19,20,21,22,23,24,25,26,27,28,29,30,31,32,33,34,35,36].

Methods Used in Ocular Microbiome Exploration	Advantages	Disadvantages
culture-based methods	- long established and well described - isolate only microbes alive	- not able to identify viruses and difficult to culture bacteria and fungi - time-consuming - lower positivity rate than sequencing - not adequate for the characterization of novel microbiomes continuously exposed to environment - not able to describe and quantify the vast composition of complex microbiomes
16SrRNA gene-based sequencing methods	- the most commonly used sequencing technology in microbiome research so far (well described) - adequate for identifying low-abundance microbiome	- detect only bacteria - do not provide absolute quantification of microbial DNA
whole-metagenome shotgun sequencing methods	- emerging sequencing technology - sequence DNA regardless of its origin - enable the detection of viruses, archaea, and eukaryotes in addition to bacteria - increase taxonomic resolution and provides deeper genomic information - high sensitivity	- high sensitivity, that may be a reason for the artifactual identification of microbial species (contaminants) - do not differentiate between detected alive and dead organisms - do not provide absolute quantification of microbial DNA

**Table 2 microorganisms-12-02232-t002:** Ocular surface microbiota composition in order of frequency and quantity [14,44,45,46,47,48,49,50,51,52].

Bacteria		Viruses	Fungi	
Phylum	Genus		Phylum	Genus
*Proteobacteria*	*Corynebacterium*	*Torque teno Virus TTV*	*Basidiomycota*	*Malassezia*
*Actinobacteria*	*Pseudomonas*	*Multiple sclerosis-associated retrovirus*	*Ascomycota*	*Rhodotorula*
*Firmicutes*	*Staphylococcus*	*Human Endogenous Retrovirus K*		*Davidiella*
*Bacteroidetes*	*Streptococcus*			*Aspergillus*
*Cyanobacteria*	*Acinetobacter*			*Alternaria*
*Deinoococcus thermus*	*Propionibacterium*			
	*Bacillus*			
	*Agrobacterium*			
	*Sphingomonas*			
	*Enhydrobacter*			

**Table 3 microorganisms-12-02232-t003:** Most common alterations of ocular microbiota composition in ocular disorders.

Disease	Increased Abundance	Decreased Abundance
Blepharitis	*Staphylococcus*,*Streptophyta*,*Corynebacterium*, *Enhydrobacter*	*Propionibacterium*
Meibomian gland dysfunction—moderate		*Staphylococcus*
Meibomian gland dysfunction—severe	*Propionibacterium acnes*	
Dry eye syndrome	Gram-positive bacteria, especially coagulase-negative *Staphylococci*,*S. aureus*, and*Corynebacterium*, andpathogens such as *Rhodococcus* and *K. oxytoca*	
Trachoma	*Corynebacterium* *Streptococcus*	
Infectious keratitis	*Pseudomonas* *Acinetobacter*

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
