# Peer review of "The Ocular Microbiome: Micro-Steps Towards Macro-Shift in Targeted Treatment? A Comprehensive Review"

_microorganisms, 2024, doi:10.3390/microorganisms12112232_

Round 1
Reviewer 1 Report
Comments and Suggestions for Authors
dear authors,
it is an interesting work and i want also to give you some ideas
Drawing from such findings, it is plausible that nanoparticle-based therapies, or other microbiome-modulating treatments, could induce similar stress responses in ocular surface cells. Therefore, further research into how therapeutic agents affect both the microbiome and the cellular phenotypes of ocular tissues is necessary to optimize treatment strategies and avoid unintended consequences such as altered cellular functions or resistance to therapy.
We reccomend to insert data about the role of nanoparticles in disrupting the oral microbiota. doi: 10.3390/ph14101007. PMID: 34681232; PMCID: PMC8537856.
Author Response
Dear Reviewer,
We are grateful for Your detailed comments on our manuscript. Please find detailed answers below and in the revised manuscript.
Reviewers` comments:
Dear authors,
it is an interesting work and I want also to give you some ideas.
Re: Thank you for your time and effort to review our manuscript. We are more than grateful for the opportunity to improve our paper with modifications you suggested.
Drawing from such findings, it is plausible that nanoparticle-based therapies, or other microbiome- modulating treatments, could induce similar stress responses in ocular surface cells. Therefore, further research into how therapeutic agents affect both the microbiome and the cellular phenotypes of ocular tissues is necessary to optimize treatment strategies and avoid unintended consequences such as altered cellular functions or resistance to therapy.
We reccomend to insert data about the role of nanoparticles in disrupting the oral microbiota. Doi: 10.3390/ph14101007. PMID: 34681232; PMCID: PMC8537856
Re: We fully agree with this remark.
We have included the information about possible consequences of new treatment modalities for ocular surface cells and microbiome, on an example of nanoparticle-based therapies as well as the need of further research of such influence and optimized treatment strategies. (lines: 403-405)
We, as suggested, included additional bibliography position: 138. Mîndrilă I, Osman A, Mîndrilă B, Predoi MC, Mihaiescu DE, Buteică SA, Phenotypic Switching of B16F10 Melanoma Cells as a Stress Adaptation Response to Fe3O4/Salicylic Acid Nanoparticle Therapy, Pharmaceuticals (Basel), 2021, 14(10):1007
Sincerely,
Joanna Przybek-Skrzypecka, MD, PhD
Corresponding author

Reviewer 2 Report
Comments and Suggestions for Authors
This article reviews available data on the ocular microbiome and microbiota, their role in maintaining ocular homeostasis, and the impact of dysbiosis on several ophthalmic disorders. Although the topic is interesting in its scientific field, there are some issues that require the authors’ attention to improve the quality of this particular manuscript before further consideration for publication in a high-quality journal “Microorganisms”.
Specific comments:
1. The authors should enrich the article content by discussing the interactions between different components of the microbiome (e.g., bacteria, fungi, and viruses) and their collective impact on eye health in detail.
2. How the ocular microbiome may affect the pathogenesis of specific eye diseases? Please justify.
3. This article presents various methods for studying the ocular microbiome. But, the authors do not discuss the advantages and disadvantages of each method in detail. Please improve this important issue by giving comparative analysis.
4. The authors should provide a more comprehensive discussion of the limitations of current research methods used for examining the ocular microbiome.
5. The authors should consider the inclusion of more figures in the manuscript to highlight the meaningful data presentation and attract the attention from the readers.
6. As stated by the authors, the pathophysiology of AMD involves recruitment of microglia and macrophages in the subretinal and choroidal areas, mast cells, and RPE immune activation [94]. Hence, investigators have developed treatment modality to alleviate AMD (DOI: 10.1021/acsnano.2c05824). In my opinion, the authors may consider the inclusion of the aforementioned relevant case study in the reference list to strengthen manuscript quality and balance scientific viewpoint. The description of this situation may reflect the current efforts made by many other investigators and have a better connection with the authors’ focus on ocular microbiome (another promising direction for curing AMD).
Author Response
Dear Reviewer,
We are grateful for Your detailed comments on our manuscript. Please find detailed answers below and in the revised manuscript.
Reviewers` comments:
General comment:
This article reviews available data on the ocular microbiome and microbiota, their role in maintaining ocular homeostasis, and the impact of dysbiosis on several ophthalmic disorders. Although the topic is interesting in its scientific field, there are some issues that require the authors’ attention to improve the quality of this particular manuscript before further consideration for publication in a high-quality journal “Microorganisms”.
Re: Thank you for your time and effort to review our manuscript. We are more than grateful for the opportunity to improve our paper with modifications you suggested.
Specific comments:
- The authors should enrich the article content by discussing the interactions between different components of the microbiome (e.g., bacteria, fungi, and viruses) and their collective impact on eye health in detail.
Re: Thank you for that comment. We have described interactions between bacteria and viruses on an example of bacteriophages and between bacteria and fungi, as well as the influence of aforementioned interplays on eye health and ocular diseases. (lines: 130-137)
additional bibliography positions:
- Kumar V, Baweja M, Singh PK, Shukla P, Recent developments in systems biology and metabolic engineering of plant–microbe interactions. Review, Front Plant Sci, 2016, 7
- Zárate S, Taboada B, Yocupicio-Monroy M, Arias CF, Human Virome, Arch Med Res, 2017, 48(8):701–716
- Shkoporov AN, Turkington CJ, Hill C, Mutualistic interplay between bacteriophages and bacteria in the human gut, Nat Rev Microbiol, 2022, 20(12):737–749
- Kortright KE, Chan BK, Koff JL, Turner PE, Phage therapy: a renewed approach to combat antibiotic-resistant bacteria, Cell Host Microbe, 2019, 25(2):219–232
- Ge C, Wei C, Yang BX, Cheng J, Huang YS, Conjunctival microbiome changes associated with fungal keratitis: metagenomic analysis, Int J Ophthalmol, 2019, 12(2):194–200
- How the ocular microbiome may affect the pathogenesis of specific eye diseases? Please justify.
Re: We fully agree with your comment. We have depicted the interactions between ocular microbiome and eye surface as well as unordinary relationship between ocular microbiome and immune system, resulting in specific conditions for development of eye diseases. (lines: 262-281, 301-311)
additional bibliography positions:
- Zannella C, Shinde S, Vitiello M, Falanga A, Galdiero E, Fahmi A, Santella B, Nucci L, Gasparro R, Galdiero M, Boccellino M, Franci G, Di Domenico M, Antibacterial Activity of Indolicidin-Coated Silver Nanoparticles in Oral Disease, Appl Sci, 2020, 10:1837
- Caspi RR, In this issue: Immunology of the eye-inside and out, Int Rev Immunol, 2013, 32:1–3
- Kumar A, Yu FS, Toll-like receptors and corneal innate immunity, Curr Mol Med, 2006, 6:327–337
- Akira S, Takeda K, Kaisho T, Toll-like receptors: Critical proteins linking innate and acquired immunity, Nat Immunol, 2001, 2:675–680
- Pearlman E, Johnson A, Adhikary G, Sun Y, Chinnery HR, Fox T, Kester M, McMenamin PG, Toll-like receptors at the ocular surface, Ocul Surf, 2008, 6:108–116
- Ueta M, Kinoshita S, Innate immunity of the ocular surface, Brain Res Bull, 2010, 81:219–228
- Ueta M, Nochi T, Jang MH, Park EJ, Igarashi O, Hino A, Kawasaki S, Shikina T, Hiroi T, Kinoshita S, Kiyono H, Intracellularly expressed TLR2s and TLR4s contribution to an immunosilent environment at the ocular mucosal epithelium, J Immunol, 2004, 173:3337–3347
- Johnson AC, Heinzel FP, Diaconu E, Sun Y, Hise AG, Golenbock D, Lass JH, Pearlman E, Activation of toll-like receptor (TLR)2, TLR4, and TLR9 in the mammalian cornea induces MyD88-dependent corneal inflammation, Investig Ophthalmol Vis Sci, 2005, 46:589–595
- Horai R, Zárate-Bladés CR, Dillenburg-Pilla P, Chen J, Kielczewski JL, Silver PB, Jittayasothorn Y, Chan C-C, Yamane H, Honda K, Caspi RR, Microbiota-dependent activation of an autoreactive T cell receptor provokes autoimmunity in an immunologically privileged site, Immunity, 2015, 43(2):343–353
- Kugadas A, Gadjeva M, Impact of microbiome on ocular health, Ocul Surf, 2016, 14(3):342–349
- Song J, Dong H, Wang T, Yu H, Yu J, Ma S, Song X, Sun Q, Xu Y, Liu M, What is the impact of microbiota on dry eye: a literature review of the gut-eye axis, BMC Ophthalmol, 2024, 24(1):262
- Zhou L, Huang LQ, Beuerman RW, Grigg ME, Li SFY, Chew FT, Ang L, Stern ME, Tan D, Proteomic analysis of human tears: defensin expression after ocular surface surgery, J Proteome Res, 2004, 3(3):410–416
- Nakatsukasa M, Sotozono C, Shimbo K, Ono N, Miyano H, Okano A, Hamuro J, Kinoshita S, Amino Acid profiles in human tear fluids analyzed by high-performance liquid chromatography and electrospray ionization tandem mass spectrometry, Am J Ophthalmol, 2011, 151(5):799–808 e1
- Rusciano D, Roszkowska AM, Gagliano C, Pezzino S, Free amino acids: an innovative treatment for ocular surface disease, Eur J Pharmacol, 2016, 787:9–19
- This article presents various methods for studying the ocular microbiome. But, the authors do not discuss the advantages and disadvantages of each method in detail. Please improve this important issue by giving comparative analysis.
Re: Thank you for that comment. We have expanded the information about advantages, disadvantages and limitations of research methods currently used in examining the ocular microbiome. Table 1 depicts the summary of our findings. We also added bibliography positions discussing aforementioned problem.
additional bibliography positions:
- Keilty RA, The bacterial flora of the normal conjunctiva with comparative nasal culture study, Am J Ophthalmol,1930, 13(10):876–879
- Nolan J, Evaluation of conjunctival and nasal bacterial cultures before intra-ocular operations, Br J Ophthalmol, 1967, 51(7):483–485
- Perkins RE, Kundsin RB, Pratt MV, Abrahamsen I, Leibowitz HM, Bacteriology of normal and infected conjunctiva, J Clin Microbiol, 1975, 1(2):147–149
- McNatt J, Allen SD, Wilson LA, Dowell VR, Anaerobic flora of the normal human conjunctival sac, Arch Ophthalmol, 1978, 96(8):1448–1450
- Hahn MW, Koll U, Schmidt J, Isolation and cultivation of bacteria. In: The Structure and Function of Aquatic Microbial Communities, Springer, 2019:313–351
- Pham VH, Kim J. Cultivation of unculturable soil bacteria. Trends Biotechnol. 2012;30(9):475–484.
- Brooks JP, Edwards DJ, Harwich MD, Rivera MC, Fettweis JM, Serrano MG, Reris RA, Sheth NU, Huang B, Girerd P, Vaginal Microbiome Consortium, Strauss JF 3rd, Jefferson KK, Buck GA, The truth about metagenomics: quantifying and counteracting bias in 16S rRNA studies, BMC Microbiol, 2015,15:66
- Kennedy K, Hall MW, Lynch MD, Moreno-Hagelsieb G, Neufeld JD, Wommack KE, Evaluating bias of illumina-based bacterial 16S rRNA gene profiles, Appl Environ Microbiol, 2014, 80(18):5717–5722
- Costea PI, Zeller G, Sunagawa S, Pelletier E, Alberti AA, Levenez F, Tramontano M, Driessen M, Hercog R, Jung FE, Kultima JR, Hayward MR, Coelho LP, Allen-Vercoe E, Bertrand L, Blaut M, Brown JRM, Carton T, Cools-Portier S, Daigneault M, Derrien M, Druesne A, De Vos WM, Finlay BB, Flint HJ, Guarner F, Hattori M, Heilig H, Luna RA, van Hylckama Vlieg J, Junick J, Klymiuk I, Langella P, Le Chatelier E, Mai V, Manichanh C, Martin JC, Mery C, Morita H, O'Toole PW, Orvain C, Patil KR, Penders J, Persson S, Pons N, Popova M, Salonen A, Saulnier D, Scott KP, Singh B, Slezak K, Veiga P, Versalovic J, Zhao L, Zoetendal EG, Ehrlich SD, Dore J, Bork P, Towards standards for human fecal sample processing in metagenomic studies, Nat Biotechnol, 2017, 35(11):1069–1076
- Riesenfeld CS, Schloss PD, Handelsman J, Metagenomics: genomic analysis of microbial communities, Annu Rev Genet, 2004, 38:525–552
- Salter SJ, Cox MJ, Turek EM, Calus ST, Cookson WO, Moffatt MF, Turner P, Parkhill J, Loman NJ, Walker AW, Reagent and laboratory contamination can critically impact sequence-based microbiome analyses, BMC Biol, 2014, 12(1):1–12
- Tettamanti Boshier FA, Srinivasan S, Lopez A, Hoffman NG, Proll S, Fredricks DN, Schiffer JT, Complementing 16S rRNA gene amplicon sequencing with total bacterial load to infer absolute species concentrations in the vaginal microbiome. Msystems, 2020, 5(2)
- Hanson B, Zhou Y, Bautista EJ, Urch B, Speck M, Silverman F, Muilenberg M, Phipatanakul W, Weinstock G, Sodergren E, Gold DR, Sordillo JE, Characterization of the bacterial and fungal microbiome in indoor dust and outdoor air samples: a pilot study, Environ Sci Process Impacts, 2016, 18(6):713–724
- The authors should provide a more comprehensive discussion of the limitations of current research methods used for examining the ocular microbiome.
Re: We have expanded the information about advantages, disadvantages and limitations of research methods currently used in examining the ocular microbiome. We added another table- Table 1 depicts limitations of current research methods applied in ocular microbiome examination. We also added some bibliography positions to further evaluate the problem mentioned:
- Keilty RA, The bacterial flora of the normal conjunctiva with comparative nasal culture study, Am J Ophthalmol,1930, 13(10):876–879
- Nolan J, Evaluation of conjunctival and nasal bacterial cultures before intra-ocular operations, Br J Ophthalmol, 1967, 51(7):483–485
- Perkins RE, Kundsin RB, Pratt MV, Abrahamsen I, Leibowitz HM, Bacteriology of normal and infected conjunctiva, J Clin Microbiol, 1975, 1(2):147–149
- McNatt J, Allen SD, Wilson LA, Dowell VR, Anaerobic flora of the normal human conjunctival sac, Arch Ophthalmol, 1978, 96(8):1448–1450
- Hahn MW, Koll U, Schmidt J, Isolation and cultivation of bacteria. In: The Structure and Function of Aquatic Microbial Communities, Springer, 2019:313–351
- Pham VH, Kim J. Cultivation of unculturable soil bacteria. Trends Biotechnol. 2012;30(9):475–484.
- Brooks JP, Edwards DJ, Harwich MD, Rivera MC, Fettweis JM, Serrano MG, Reris RA, Sheth NU, Huang B, Girerd P, Vaginal Microbiome Consortium, Strauss JF 3rd, Jefferson KK, Buck GA, The truth about metagenomics: quantifying and counteracting bias in 16S rRNA studies, BMC Microbiol, 2015,15:66
- Kennedy K, Hall MW, Lynch MD, Moreno-Hagelsieb G, Neufeld JD, Wommack KE, Evaluating bias of illumina-based bacterial 16S rRNA gene profiles, Appl Environ Microbiol, 2014, 80(18):5717–5722
- Costea PI, Zeller G, Sunagawa S, Pelletier E, Alberti AA, Levenez F, Tramontano M, Driessen M, Hercog R, Jung FE, Kultima JR, Hayward MR, Coelho LP, Allen-Vercoe E, Bertrand L, Blaut M, Brown JRM, Carton T, Cools-Portier S, Daigneault M, Derrien M, Druesne A, De Vos WM, Finlay BB, Flint HJ, Guarner F, Hattori M, Heilig H, Luna RA, van Hylckama Vlieg J, Junick J, Klymiuk I, Langella P, Le Chatelier E, Mai V, Manichanh C, Martin JC, Mery C, Morita H, O'Toole PW, Orvain C, Patil KR, Penders J, Persson S, Pons N, Popova M, Salonen A, Saulnier D, Scott KP, Singh B, Slezak K, Veiga P, Versalovic J, Zhao L, Zoetendal EG, Ehrlich SD, Dore J, Bork P, Towards standards for human fecal sample processing in metagenomic studies, Nat Biotechnol, 2017, 35(11):1069–1076
- Riesenfeld CS, Schloss PD, Handelsman J, Metagenomics: genomic analysis of microbial communities, Annu Rev Genet, 2004, 38:525–552
- Salter SJ, Cox MJ, Turek EM, Calus ST, Cookson WO, Moffatt MF, Turner P, Parkhill J, Loman NJ, Walker AW, Reagent and laboratory contamination can critically impact sequence-based microbiome analyses, BMC Biol, 2014, 12(1):1–12
- Tettamanti Boshier FA, Srinivasan S, Lopez A, Hoffman NG, Proll S, Fredricks DN, Schiffer JT, Complementing 16S rRNA gene amplicon sequencing with total bacterial load to infer absolute species concentrations in the vaginal microbiome. Msystems, 2020, 5(2)
- Hanson B, Zhou Y, Bautista EJ, Urch B, Speck M, Silverman F, Muilenberg M, Phipatanakul W, Weinstock G, Sodergren E, Gold DR, Sordillo JE, Characterization of the bacterial and fungal microbiome in indoor dust and outdoor air samples: a pilot study, Environ Sci Process Impacts, 2016, 18(6):713–724
- The authors should consider the inclusion of more figures in the manuscript to highlight the meaningful data presentation and attract the attention from the readers.
Re: Thank you for that comment. We have expanded the information about advantages, disadvantages and limitations of research methods currently used in examining the ocular microbiome.
To simplify and make the manuscript clearer and attractive, for this comparative analysis we have used the table form (Table 1).
additional bibliography positions:
- Keilty RA, The bacterial flora of the normal conjunctiva with comparative nasal culture study, Am J Ophthalmol,1930, 13(10):876–879
- Nolan J, Evaluation of conjunctival and nasal bacterial cultures before intra-ocular operations, Br J Ophthalmol, 1967, 51(7):483–485
- Perkins RE, Kundsin RB, Pratt MV, Abrahamsen I, Leibowitz HM, Bacteriology of normal and infected conjunctiva, J Clin Microbiol, 1975, 1(2):147–149
- McNatt J, Allen SD, Wilson LA, Dowell VR, Anaerobic flora of the normal human conjunctival sac, Arch Ophthalmol, 1978, 96(8):1448–1450
- Hahn MW, Koll U, Schmidt J, Isolation and cultivation of bacteria. In: The Structure and Function of Aquatic Microbial Communities, Springer, 2019:313–351
- Pham VH, Kim J. Cultivation of unculturable soil bacteria. Trends Biotechnol. 2012;30(9):475–484.
- Brooks JP, Edwards DJ, Harwich MD, Rivera MC, Fettweis JM, Serrano MG, Reris RA, Sheth NU, Huang B, Girerd P, Vaginal Microbiome Consortium, Strauss JF 3rd, Jefferson KK, Buck GA, The truth about metagenomics: quantifying and counteracting bias in 16S rRNA studies, BMC Microbiol, 2015,15:66
- Kennedy K, Hall MW, Lynch MD, Moreno-Hagelsieb G, Neufeld JD, Wommack KE, Evaluating bias of illumina-based bacterial 16S rRNA gene profiles, Appl Environ Microbiol, 2014, 80(18):5717–5722
- Costea PI, Zeller G, Sunagawa S, Pelletier E, Alberti AA, Levenez F, Tramontano M, Driessen M, Hercog R, Jung FE, Kultima JR, Hayward MR, Coelho LP, Allen-Vercoe E, Bertrand L, Blaut M, Brown JRM, Carton T, Cools-Portier S, Daigneault M, Derrien M, Druesne A, De Vos WM, Finlay BB, Flint HJ, Guarner F, Hattori M, Heilig H, Luna RA, van Hylckama Vlieg J, Junick J, Klymiuk I, Langella P, Le Chatelier E, Mai V, Manichanh C, Martin JC, Mery C, Morita H, O'Toole PW, Orvain C, Patil KR, Penders J, Persson S, Pons N, Popova M, Salonen A, Saulnier D, Scott KP, Singh B, Slezak K, Veiga P, Versalovic J, Zhao L, Zoetendal EG, Ehrlich SD, Dore J, Bork P, Towards standards for human fecal sample processing in metagenomic studies, Nat Biotechnol, 2017, 35(11):1069–1076
- Riesenfeld CS, Schloss PD, Handelsman J, Metagenomics: genomic analysis of microbial communities, Annu Rev Genet, 2004, 38:525–552
- Salter SJ, Cox MJ, Turek EM, Calus ST, Cookson WO, Moffatt MF, Turner P, Parkhill J, Loman NJ, Walker AW, Reagent and laboratory contamination can critically impact sequence-based microbiome analyses, BMC Biol, 2014, 12(1):1–12
- Tettamanti Boshier FA, Srinivasan S, Lopez A, Hoffman NG, Proll S, Fredricks DN, Schiffer JT, Complementing 16S rRNA gene amplicon sequencing with total bacterial load to infer absolute species concentrations in the vaginal microbiome. Msystems, 2020, 5(2)
- Hanson B, Zhou Y, Bautista EJ, Urch B, Speck M, Silverman F, Muilenberg M, Phipatanakul W, Weinstock G, Sodergren E, Gold DR, Sordillo JE, Characterization of the bacterial and fungal microbiome in indoor dust and outdoor air samples: a pilot study, Environ Sci Process Impacts, 2016, 18(6):713–724
- As stated by the authors, the pathophysiology of AMD involves recruitment of microglia and macrophages in the subretinal and choroidal areas, mast cells, and RPE immune activation [94]. Hence, investigators have developed treatment modality to alleviate AMD (DOI: 10.1021/acsnano.2c05824). In my opinion, the authors may consider the inclusion of the
aforementioned relevant case study in the reference list to strengthen manuscript quality and balance scientific viewpoint. The description of this situation may reflect the current efforts made by many other investigators and have a better connection with the authors’ focus on ocular microbiome (another promising direction for curing AMD).
Re: We fully agree that it is worth to mention the innovative modality of AMD treatment, described in suggested study. We have included the description of investigated by Nguyen et al. nanotherapeutic, as an example of current efforts to find new direction for effective AMD treatment. (lines: 395-402)
additional bibliography positions: 137. Nguyen DD, Luo LJ, Yang CJ, Lai JY, Highly Retina-Permeating and Long-Acting Resveratrol/Metformin Nanotherapeutics for Enhanced Treatment of Macular Degeneration, ACS Nano, 2023, 17(1):168-183
Sincerely,
Joanna Przybek-Skrzypecka, MD, PhD
Corresponding author

Round 2
Reviewer 2 Report
Comments and Suggestions for Authors
The revised version has adequately addressed most of the critiques raised by this reviewer and is now suitable for publication in "Microorganisms".